# OpenReview forum: "Uncertainty Quantification in SVM prediction"
_TMLR — Rejected by TMLR_

### Review · Reviewer_Gum7 · 2025-06-25

**Summary Of Contributions:**

This paper first provided seven desirable properties of an ideal prediction interval estimation by SVM as follows:

* Distribution-free method
* Asymptotic guarantees
* Direct PI estimation
* PI tube movements
* Global Optimal Solution
* Re-calibration
* Sparsity

Based on these properties, this paper replaced the L2 regularization term in SVQR with an L1 regularization term and proposed SSVQR. This replacement changed the optimization problem in the SVM from a quadratic programming problem to a linear programming problem, and the authors claimed that its solution became sparse. In numerical experiments, they compared SVQR and SSVQR from the perspective of sparsity. They obtained more sparse solutions under various conditions, while SSVQR achieved comparable prediction interval coverage probability and interval width with SVQR. They also conducted experiments examining the characteristics of SSVAR from the perspective of feature selection for high-dimensional data. Feature selection based on SSVQR improved prediction interval coverage probability and interval width and reduced the computational cost for training. Finally, for several time series prediction tasks, they compared deep learning-based conformal regression with SSVQR-based conformal regression from the viewpoint of the stability of the solution and the computational cost. They confirmed that SSVQR provided more stable solutions and reduced computational cost while achieving performance equivalent to neural networks in terms of prediction interval coverage probability and interval width.

**Audience:**

Yes

**Broader Impact Concerns:**

Not applicable.

**Claims And Evidence:**

No

**Requested Changes:**

### Major

* Please clarify the problem to be solved and the assumed situations in this paper
  * The lack of clarity about the problem to be solved causes the following confusion:
    * The introduction devotes considerable space to describing how neural networks are widely used for prediction interval estimation and that they can be divided into two classes. This suggests that the problem to be solved in this paper has characteristics suitable for neural networks, such as abundant data, rich computational resources, and strong nonlinearity. If the problem doesn't fall into these categories—for example, if you want to handle small tabular data as mentioned in the latter part of the introduction—then there's no need to describe neural network research so carefully, and comparative experiments with neural networks wouldn't necessarily be required. However, this research ultimately conducts comparative experiments with neural networks.
    * Section 3 describes desirable properties for prediction interval estimation by SVM, but whether these properties are desirable should depend on the problem and situation being addressed. In fact, the paper excludes the methods that cannot achieve the Global Optimal Solution from experimental comparisons. This implicitly suggests that this paper deals with problems where other properties aren't necessarily required if the Global Optimal Solution can be achieved. If so, there wouldn't be a need to explain other properties in such depth. Moreover, Section 4 conducts performance verification from the perspective of feature selection for high-dimensional data, which isn't included in the desirable properties.
    * If solution sparsity is particularly emphasized in this paper, then comparison with methods like (Anand et al., 2020; Ye et al., 2025) mentioned at the beginning of Section 4 should be conducted carefully. However, these methods are not discussed in Section 3, and no comparisons are made in numerical experiments.
    * In comparative experiments with neural networks, prediction interval estimation using conformal regression is performed, but conformal regression isn't used in comparative experiments with conventional SVM in Subsection 5.5.
  * In my opinion, all this confusion stems from the lack of clarity about the problem to be solved in this paper and the clarity about assumed situations. To fundamentally resolve this issue, structural changes in the manuscript and possibly revision of numerical experiments may be necessary. Personally, I believe acceptance in this submission is difficult, and major revision followed by resubmission should be considered.
* Please clearly state the definition of sparsity
  * In conventional SVM, sparsity means having few support vectors that determine the prediction function, which is achieved by using the ε-insensitive loss function. On the other hand, Subsection 4.1 states that sparsity similar to LASSO regression occurs in the solution. Sparsity in LASSO regression refers to most regression coefficients becoming zero. These are different concepts, so please clearly state the definition of sparsity in this paper.
* Please explain the derivation of the linear programing problem
  * Please add proof for derivation of Eq. (17) from Eq. (16), or cite a previous research proving this. The derivation of the dual problem in conventional SVM is based on the fact that the regularization term is an L2 regularization term (for example, the textbook by Bishop (2006) performs such derivation), and it's not obvious that similar transformations can be applied when using L1 regularization terms.
* Please explain the difference between feature selection by Algorithm 3 from the ordinary LASSO regression
  * I believe SVM using linear kernels is equivalent to linear regression models with regularization terms. What's the difference between directly solving quantile regression with regularized linear regression models and solving the dual problem with Algorithm 3 followed by feature selection? What are the advantages of using Algorithm 3?
* Please explain why feature selection reduces training time.
  * In Algorithm 3, training is performed in lines 3-6, and feature selection is performed in lines 7-8. If training time is considered as the time required for processing lines 3-6, then whether feature selection is performed shouldn't affect the training time. What does training time refer to in Subsection 5.4, and why is it reduced by feature selection?

### Minor

* In the training set $T = \\{ (x_i, y_i) : x_i \in \boldsymbol{R}^n, y_i \in \boldsymbol{R}, i = 1, 2, \dots , m. \\}$, the period after $m$ is unnecessary.
* The abbreviation HQ is not defined.
* The description in (d) on page 3, "we reveal the effectiveness of the Sparse SVM-based PI model by performing feature selection using sparse SVM PI based feature selection algorithm," seems redundant.
* There are many unnecessary hyphens such as "Section-3" and ":-".
* In Subsection 2.2, there are no references, so some papers should be cited.
* Since both the transpose symbol and training set symbol use $T$, consider using $\top$ (\\top) for the transpose symbol.
* In Subsection 2.4, "$(1-\alpha-q)^{th}$ quantiles" should be "$(1-\alpha+q)^{th}$ quantiles".
* In the text above Eq. (11), "$Q_{1-\alpha} (E_i, I_2)$" should be "$Q_{1-\alpha} (E, I_2)$".
* In line 4 of Subsection 3.4, "functions of 13" should be "functions of (13)" (missing parentheses).
* In Table 2, CP at SSVQR's third row is missing parentheses.
* In line 2 of Subsection 5.3, "rightmost column" should be "leftmost column".
* In line 3 of Subsection 5.5, "Table 10 and 13 contains" should be "Tables 10 and 11 contain".
* In Table 12, SSVQR's MPIW for Beer Prod. being 0.94 is probably incorrect.
* The following description regarding Table 14 seems inconsistent with Table 14's content:
  * First, SVQR+CR achieved the target coverage in 4 out of 5 datasets, while CQR-NN did so in only 3 out of 5. Moreover, SVQR+CR yielded lower MPIW in 4 out of 5 cases.

**Strengths And Weaknesses:**

### Strengths

* By replacing the L2 regularization term in SVQR with an L1 regularization term, the optimization problem changes from a quadratic programming problem to a linear programming problem that is easier to solve.
* In numerical experiments, improved solution sparsity is confirmed while achieving performance equivalent to standard SVQR in terms of prediction interval coverage probability and interval width.
* For several time series prediction tasks, computational cost reduction is achieved while maintaining performance equivalent to neural networks in terms of prediction interval coverage probability and interval width.

### Weaknesses

* Some descriptions are unclear.
  * The problem to be solved or the assumed situation in this paper is not clearly stated. Therefore, the algorithm's motivation and the purpose of numerical experiments are unclear.
  * The definition of sparsity in this paper is not clearly stated.
  * Insufficient explanation regarding the derivation of SSVQR's linear programming problem.

---

> ### Author Response · Authors · 2025-07-20
>
> We sincerely thank you for your kind appreciation of our work. Your thoughtful and constructive suggestions have been invaluable in enhancing the overall quality of the manuscript. In response to your comments and queries, we have carefully prepared a detailed point-by-point reply. As part of the supplementary material, we have included a single PDF file that first presents our responses to the reviewers’ comments, followed by the revised manuscript with all changes highlighted in blue.

---

### Review · Reviewer_qupH · 2025-06-26

**Summary Of Contributions:**

This paper focuses on support vector quantile regression (SVQR). It starts with a review of the basics of quantile regression and a comparison of existing methods (Sections 2 and 3). The main technical idea is to introduce a sparse version of SVQR by replacing the $L_2$-norm regularizer in SVQR with an $L_1$-norm, and to apply it to a feature selection algorithm for linear prediction interval estimation. Separately, the paper also studies how SVQR performs in the conformal regression setting, with some empirical evaluations.

**Audience:**

No

**Claims And Evidence:**

No

**Requested Changes:**

- Please clarify the motivation for focusing on the SVM setting as opposed to neural networks. While the paper emphasizes that SVQR offers explicit problem formulations, it does not demonstrate any particular theoretical advantage. For example, the paper still cites results from the general non-parametric settings (e.g., Takeuchi et al. 2006) to justify guarantees in the specific SVM setting. The authors should better explain the unique technical or practical benefits of studying SVQR in this setting.

- In Section 5.7, more details about the neural network architecture and training settings are needed to make fair comparisons.

**Strengths And Weaknesses:**

Strength
- The paper studies uncertainty quantification, which is a key topic for producing reliable machine learning predictions.

- It provides a helpful overview of the basics of quantile regression.

Weakness

- The idea of using sparse SVQR is quite straightforward and has been explored previously, e.g., in Li et al., "$L_1$-Norm Quantile Regression" (2008).

- There are no original theoretical results. The paper does not provide new proofs or derivations; instead, it relies heavily on results from prior work (e.g., Section 4.1 cites asymptotic guarantees from Takeuchi et al., 2006).

- The feature selection method is limited to linear models.

- There is no clear conceptual or technical connection between the sparse SVQR part (Section 4.1) and the conformal regression part (Section 4.3). As a result, the paper reads more like a collection of loosely related ideas than a cohesive study with a clear central contribution.

- In Section 5.7, the authors claim that SVQR+CR outperforms CQR-NN, but the comparison is unconvincing. More details about the neural network architecture and training settings are necessary to make fair comparisons.

---

> ### Author Response · Authors · 2025-07-20
>
> Authors are thankful for your comment. You comment has improved the flow and content of our manuscript. As per your question, we have revised our manuscript. In response to your comments and queries, we have prepared a detailed point-by-point reply. We have provided the single pdf file in the supplementary material that first presents our point-by-point responses to reviewer’ comments, followed by the revised manuscript with all changes highlighted in blue.

---

### Review · Reviewer_uh1U · 2025-06-28

**Summary Of Contributions:**

The manuscript presents an extensive study on uncertainty quantification (UQ) within the Support Vector Machine (SVM) framework, particularly targeting regression and forecasting problems. The authors propose a novel Sparse Support Vector Quantile Regression (SSVQR) model, alongside a feature selection algorithm tailored for prediction interval (PI) estimation. The work also includes an extension to conformal prediction and provides a comparative evaluation against both traditional SVM-based PI methods and modern deep learning approaches.

**Audience:**

Yes

**Broader Impact Concerns:**

no concerns

**Claims And Evidence:**

Yes

**Requested Changes:**

Major Comments

1. Clarify Experimental Details in Abstract and Main Text
    The abstract makes the claim:
 "SVM models show comparable or superior performance to modern complex deep learning models for probabilistic forecasting task in our experiments."
However, there is insufficient detail in the abstract and in some sections of the paper about the experimental setup—e.g., the nature of the datasets, evaluation metrics, and key baselines. Please add more specifics (e.g., number and type of datasets, comparison metrics) both in the abstract and the body of the paper to contextualize this claim.

2. Distinguish Clearly Between Contributions and Prior Work
The paper outlines contributions early on, but in some technical sections (notably 2.2 and 4.1), it becomes unclear whether the material represents novel contributions or reviews of prior work.
In Section 2.2 ("Support Vector Quantile Regression Model") and Section 4.1 ("Sparse Support Vector Quantile Regression Model"), clearly indicate which parts are background and which are proposed methods.

3. Discussion on Linear Programming Solvers (LPP)
Since the proposed method relies on solving a pair of linear programs, it would be valuable to include a short discussion on:
What solvers are used (open-source alternatives,...).
Computational complexity and scalability.
Any known limitations or performance bottlenecks.
This will help readers assess the practicality of applying SSVQR to larger-scale problems.

4. Improve Figures and Tables
Many figures and tables could benefit from higher quality and better formatting:

Figure 3: Low image resolution makes it hard to interpret.

Figure 4: Image quality is poor; also, panel (a) has an aspect ratio issue.

Figure 5: The x-axis label includes a misplaced "q".

Figure 6: Image quality is low; some text is difficult to read.

Table 14: Formatting could be improved for readability.

Please consider increasing resolution, standardizing font sizes, and ensuring all labels are legible.

5. Highlight Best Results in Tables
    As many results are presented in tabular form, it would be helpful to bold the best-performing results for each metric and dataset. This would greatly enhance readability and help readers quickly identify key outcomes.

**Strengths And Weaknesses:**

Strengths

- The paper addresses a relevant and underexplored topic: UQ for SVM models, which are often underutilized in probabilistic forecasting.

- The proposed SSVQR method introduces sparsity—a hallmark of SVM models—into quantile regression for UQ, which is novel and well-motivated.
- The inclusion of a feature selection mechanism for PI estimation in high-dimensional settings is valuable.

- The extension to conformal prediction further strengthens the practical utility of the method.

- The authors present thorough experimental comparisons across multiple datasets and methods.

---

> ### Author Response · Authors · 2025-07-20
>
> First of all, the authors thank you for appreciating our work. Your valuable suggestions have significantly contributed to improving the overall quality of the manuscript. In response to your comments and queries, we have prepared a detailed point-by-point reply. We have provided the single pdf file in the supplementary material that first presents our point-by-point responses to reviewer’ comments, followed by the revised manuscript with all changes highlighted in blue.

---

### Comment · Action_Editor_hBMK · 2025-07-04
**Updated Paper**

Dear all,

After communicating with the authors, I granted them an extension until July 20 to answer all the reviewers' comments, and upload an updated version of the manuscript.

Thank you to the reviewers for their hard work and very detailed reviews,

The AE

---

> ### Author Response · Authors · 2025-07-20
>
> Thank you for giving us the opportunity to revise our manuscript. The reviewer comments have been instrumental in enhancing the overall quality of our work. We have carefully revised the manuscript to incorporate the reviewers' suggestions and have provided a detailed point-by-point response to all queries and concerns. As part of the supplementary material, we have included a single PDF that first presents our point-by-point responses to the reviewers’ comments, followed by the revised manuscript with all changes highlighted in blue.
>
>
> As per the reviewer queries, we have clearly detailed the motivation behind the studying the SVM models from the lens of the UQ in the revised manuscript and also further argues that why the SVM models are less uncertain, more stable and trustworthy than Neural Network (NN). We have modified the contributions and abstract in the revised work for improving the overall  comprehension to the reader. We have also modified the Conclusion Section to reflect the impact of our work. Additionally, we have clearly defined our experimental objectives in starting of the Experimental Section and systematically presented the numerical results, including details of the experimental setup, parameter tuning, and baseline models. Furthermore, we have also added a detailed mathematical derivation of the SSVQR optimization problem.
>
> Motivation:-
>
> The UQ tasks in regression setting such as PI estimation, probabilistic forecasting and conformal regression derive a pair of quantile functions to quantify the uncertainty in the relationship between $X$ and $Y$,  However, their estimates themselves may also involve a certain degree of uncertainty. Therefore, researchers typically consider two main sources of uncertainty when assessing the overall uncertainty: data uncertainty (aleatoric uncertainty) and model uncertainty (epistemic uncertainty). Data uncertainty arises from the inherent noise or variability in the relationship between $X$ and $Y$ while model uncertainty primarily arises from the uncertainty in the estimated model parameters.
>
> For UQ task in NN and deep learning regression, accurately capturing model uncertainty, in addition to data uncertainty is essential.
> This is due to the significant variability in the learned weights, as the underlying objective functions are typically non-convex. As a result, training the same model multiple times with the same dataset and hyperparameter settings can still lead to different local minima. Studies by (Lakshminarayanan et al., 2017) and (Pearce et al., 2018) have shown that ensemble methods are among the most effective approaches for addressing model uncertainty in NN. The key idea is to train the model multiple times with different random initializations on the same training set $T$, and then estimate model uncertainty by measuring the variability in the resulting learned parameters.
>
> However, in contrast to NN, SVM does not suffer from such parameter uncertainty, as they obtain the global optimal solution that remains invariant to the choice of the initialization. It makes the SVM \textbf{more trustworthy and less uncertain} than NN. Apart from the global optima, the SVM also offers simple,  interpretable and often sparse solutions along with an explicit mechanism to incorporate the regularization.
>
>
> Despite these notable advantages including the zero parameter uncertainty, SVM remains almost underexplored from the lens of the UQ. In contrast to detailed NN literature, only a limited number of SVM-based approaches have been proposed for PI estimation and probabilistic forecasting tasks. Moreover, SVM models have yet to be extended into the conformal setting for the regression task. Our work addresses these gaps in the literature by extending contemporary UQ techniques to the SVM framework, supported by a comprehensive analysis and comparisons.

---

> > ### Author Response · Authors · 2025-07-21
> >
> > Dear Editor,
> > I have attached a single PDF as supplementary material, which first presents our point-by-point responses to the reviewers’ comments, followed by the revised manuscript with all changes highlighted in blue. As the reviews and corresponding responses are quite extensive, I have not posted individual replies in the comment box on the portal. I hope this is acceptable. Also ask me if I need to post them explicitly in the comment boxes.
> > Thanks
> > Pritam.

---

### Decision · Action_Editor_hBMK · 2025-08-03

**Recommendation:** Reject

**Additional Comments:**

The authors can resubmit once they address the problems listed above. They can also repackage the paper as a survey on SVQR.

**Audience:**

Yes

**Audience Explanation:**

This paper can be casted as a review paper for SVQR for a general audience.

**Claims And Evidence:**

No

**Claims Explanation:**

The reviewers express several reservations regarding the content of the paper. They can be summarized as follows.

1) Despite being pointed out in the Requested Changes, the term "sparsity" is not clearly defined, and different concepts are expressed using the same term "sparsity." This makes the paper very difficult to understand and may lead to misunderstandings by readers. It seems that the meaning depends on the nature of applications.

2) The explanation for the derivation from equation (16) to equation (17) is insufficient. The reference (Takeuchi et al., 2006) cited by the authors does not contain descriptions that directly prove the authors' claim.

3) What Algorithm 3 is solving is essentially not an SVM but merely linear quantile regression with L1 regularization. Therefore, the solved problem is essentially the same as the problem in (Li et al., 2008).

**Resubmission Of Major Revision:**

The authors may consider submitting a major revision at a later time.